

# The Flux-Anomaly-Forced Model Intercomparison Project (FAFMIP) contribution to CMIP6: Investigation of sea-level and ocean climate change in response to $CO_2$ forcing

Jonathan M. Gregory[1,2], Nathaelle Bouttes-Mauhourat[3], Stephen M. Griffies[4], Helmuth Haak[5], William J. Hurlin[4], Johann Jungclaus[5], Maxwell Kelley[6], Warren G. Lee[7], John Marshall[8], Anastasia Romanou[6], Oleg A. Saenko[7], Detlef Stammer[9], and Michael Winton[4]

[1]NCAS, University of Reading, UK
[2]Met Office Hadley Centre, Exeter, UK
[3]University of Bordeaux, France
[4]NOAA Geophysical Fluid Dynamics Laboratory, Princeton, USA
[5]Max Planck Institute for Meteorology, Hamburg, Germany
[6]Goddard Institute for Space Sciences, Columbia University, New York, USA
[7]Canadian Centre for Climate Modelling and Analysis, Victoria, British Columbia, Canada
[8]Department of Earth, Atmospheric and Planetary Sciences, Massachusetts Institute of Technology, Cambridge, USA
[9]University of Hamburg, Germany
*Correspondence to:* J. M. Gregory (j.m.gregory@reading.ac.uk)

**Abstract.** The Flux-Anomaly-Forced Model Intercomparison Project (FAFMIP) aims to investigate the spread in simulations of sea-level and ocean climate change in response to $CO_2$ forcing by atmosphere-ocean general circulation models (AOGCMs). It is particularly motivated by the uncertainties in projections of ocean heat uptake, global-mean sea-level rise due to thermal expansion and the geographical patterns of sea-level change due to ocean density and circulation change. FAFMIP has three

tier-1 experiments, in which prescribed surface flux perturbations of momentum, heat and freshwater respectively are applied to the ocean in separate AOGCM simulations. All other conditions are as in the pre-industrial control. The prescribed fields are typical of pattern and magnitude of changes in these fluxes projected by AOGCMs for doubled $CO_2$ concentration. Five groups have tested the experimental design with existing AOGCMs. Their results show diversity in the pattern and magnitude of changes, with some common qualitative features. Heat and water flux perturbation cause the dipole in sea-level change in

the North Atlantic, while momentum and heat flux perturbation cause the gradient across the Antarctic Circumpolar Current. The Atlantic Meridional Overturning Circulation (AMOC) declines in response to the heat flux perturbation, and there is a strong positive feedback on this effect due to the consequent cooling of sea surface temperature in the North Atlantic, which enhances the local heat input to the ocean. The momentum and water flux perturbations do not substantially affect the AMOC. Heat is taken up largely as a passive tracer in the Southern Ocean, which is the region of greatest heat input, but elsewhere heat

is actively redistributed towards lower latitude. Future analysis of these and other phenomena with the wider range of CMIP6 FAFMIP AOGCMs will benefit from new diagnostics of temperature and salinity tendencies, which will enable investigation of the model spread in behaviour in terms of physical processes as formulated in the models.

**Keywords.** Sea-level, dynamic sea-level change, thermosteric sea-level change, ocean heat uptake, climate change, AMOC



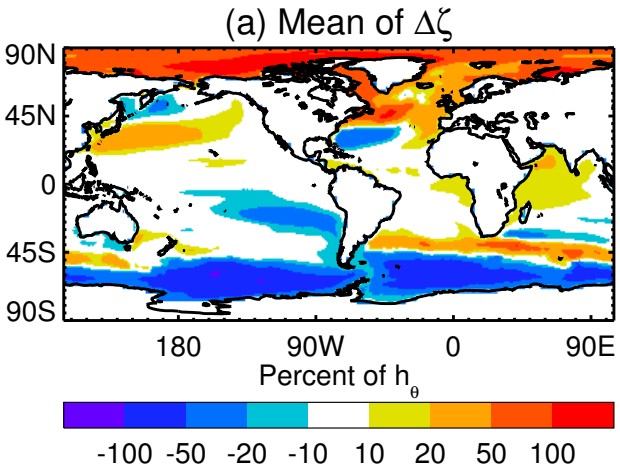
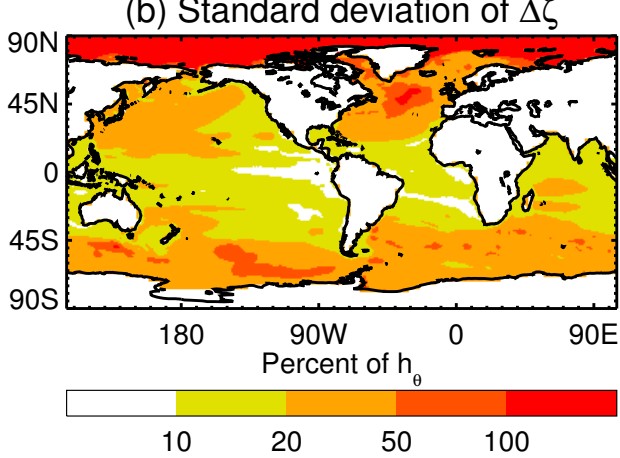

**Figure 1.** (a) Ensemble mean and (b) ensemble standard deviation of CMIP5 AOGCMs for the projected change $\Delta\zeta$ in ocean dynamic sea-level for 2081–2100 with respect to 1986–2005 under the mid-range scenario RCP4.5, expressed as percentages of ensemble-mean global-mean sea-level rise $h_\theta$ due to thermal expansion for the same scenario.

# 1 Introduction

Atmosphere-Ocean General Circulation Models (AOGCMs) are widely used for projections of future sea-level change (*e.g.* Church et al., 2013; Slangen et al., 2014). On the basis of AOGCM results contributed to the Coupled Model Intercomparison Project Phase 5 (CMIP5), global-mean sea-level rise (GMSLR) of 0.32–0.63 m (5–95%, median 0.47 m) is projected under the mid-range RCP4.5 scenario considered in the Fifth Assessment Report of the Intergovernmental Panel on Climate Change (IPCC) (Yin, 2012; Church et al., 2013). Of this, 0.14–0.23 m (median 0.19 m) is the thermosteric contribution, due to expan-

sion of sea-water as the ocean takes up heat, representing 30–50% of the total. Other contributions to GMSLR are due mostly to loss of land ice. Glaciers worldwide give 15–40% of the total. The median projected contributions from the Greenland and Antarctic ice sheets are smaller, although the latter is the largest source of uncertainty.

    The range of the thermosteric contribution (hereafter denoted $h_\theta$) also represents a substantial uncertainty in projections of GMSLR. It arises partly from differences among models in climate sensitivity, determined by surface and atmospheric

responses to radiative forcing, and partly from differences in the ocean processes which transport heat from the surface and redistribute it in the interior of the ocean (Kuhlbrodt and Gregory, 2012; Hallberg et al., 2013; Melet and Meyssignac, 2015). The three-dimensional distribution of additional heat within the ocean affects $h_\theta$ because of the dependence of thermal expansivity on temperature and pressure, quantified by the "expansion efficiency of heat" (Russell et al., 2000; Griffies and Greatbatch, 2012; Kuhlbrodt and Gregory, 2012; Griffies et al., 2014), the ratio of $h_\theta$ to the global ocean increase in heat content. From

CMIP5 results, this ratio is 0.12 m YJ$^{-1}$, with a 90% confidence interval of 0.10–0.14 m YJ$^{-1}$ (Lorbacher et al., 2015), indi-



cating that there is an uncertainty in $h_\theta$ resulting from a given increase in ocean heat content. By contrast, redistribution of the salt content of the ocean makes a negligible contribution to GMSLR or its uncertainty.

Sea-level change is not expected to be globally uniform. Changes in ocean circulation, temperature and salinity (and hence density) alter dynamic sea-level $\zeta(\mathbf{x}, t)$, where $\mathbf{x}$ is location and $t$ time. This quantity is defined as

$$\zeta(\mathbf{x}, t) \equiv \eta(\mathbf{x}, t) - \overline{\eta(\mathbf{x}, t)}, \tag{1}$$

where $\eta$ is sea surface height relative to a surface on which the geopotential has a uniform and constant value, and the overline indicates the mean over the ocean area, so $\overline{\zeta} = 0$ by construction. Hence

$$\Delta \eta = \Delta \zeta + \Delta \overline{\eta}, \tag{2}$$

in which the last term is GMSLR. That is, the local change in sea-level $\Delta \eta$ has contributions from GMSLR and from change

in dynamic sea-level $\Delta \zeta$. The spatial standard deviation of the CMIP5 model-mean $\Delta \zeta$ is about 30% of the model-mean $h_\theta$ (Fig. 1a).

There is a substantial model spread in $\Delta \zeta$, although in some regions, notably the Arctic, the model spread is smaller than in the previous phases of CMIP considered by earlier IPCC reports (Yin, 2012; Bouttes et al., 2012; Church et al., 2013; Slangen et al., 2014). Nonetheless, the CMIP5 RCP4.5 local spread in the pattern, measured by the ensemble standard deviation of

$\zeta(\mathbf{x})$, is 30% on average of the model-mean $h_\theta$, and for example it exceeds 100% in the North Atlantic (Fig. 1b).

There are three features of $\Delta \zeta$ that the models have in common (Fig. 1a) (Gregory et al., 2001; Church et al., 2001; Lowe and Gregory, 2006; Landerer et al., 2007; Meehl et al., 2007; Yin et al., 2010; Pardaens et al., 2011; Yin, 2012; Church et al., 2013; Slangen et al., 2014; Bouttes and Gregory, 2014): a meridional contrast between a band of positive change to the north of the Antarctic Circumpolar Current (ACC) and a band of negative change to the south; a meridional dipole in the North

Atlantic, also positive to the north and negative to the south; and positive $\Delta \zeta$ in the Arctic. Although these qualitative features are robustly predicted, the affected regions have the largest model spread in $\Delta \zeta$.

The Southern Ocean feature results both from changes to the surface heat flux and from an intensification and southward shift of the westerly windstress, which strengthens the Ekman drift and tends to tilt the isopycnals (Mikolajewicz and Voss, 2000; Lowe and Gregory, 2006; Landerer et al., 2007; Frankcombe et al., 2013; Bouttes and Gregory, 2014; Kuhlbrodt et al.,

2015; Saenko et al., 2015; Marshall et al., 2015). Eddies tend to oppose the latter effect by removing available potential energy, thus partly compensating for the effect of windstress change in $\Delta \zeta$, and limiting the sensitivity of the circumpolar circulation to windstress change (Hallberg and Gnanadesikan, 2006; Böning et al., 2008; Farneti et al., 2010; Downes and Hogg, 2013; Farneti et al., 2015). Most AOGCMs used for multidecadal simulations do not resolve ocean eddies at high latitudes, so their results will depend on their parametrisations of eddy advection on isoneutral surfaces (*e.g.* Gent and McWilliams, 1990;

Griffies, 1998).

The North Atlantic feature in $\Delta \zeta$ is caused by increased ocean buoyancy at high latitudes under $CO_2$ forcing (Bouttes et al., 2014). The buoyancy increase is due to reduced heat loss and increased precipitation. As well as tending to raise sea-level, it leads to a reduction of the Atlantic Meridional Overturning Circulation (AMOC), by 0–50% by 2100 in CMIP5 AOGCMs,





depending on model and scenario (Collins et al., 2013). The circulation change causes a redistribution of properties, giving a

negative $\Delta\zeta$ in the the subtropical North Atlantic gyre. The enhanced sea-level rise in the Arctic has been attributed to increased buoyancy from reduction of salinity (Meehl et al., 2007; Griffies et al., 2014), consistent with greater precipitation and river inflow.

Bouttes et al. (2012) investigated how much of the model spread in CMIP5 $\Delta\zeta$ was caused by the AOGCMs' different projections of surface momentum flux change in response to increasing $CO_2$. They did so by computing the field of surface

windstress change simulated for doubled $CO_2$ by each CMIP5 AOGCM, and imposed these fields as perturbations in a set of experiments (one for each CMIP5 model) with the FAMOUS AOGCM (Smith et al., 2008), which is a low-resolution and consequently relatively inexpensive version of HadCM3. Bouttes et al. (2014) carried out a corresponding study for surface heat flux and freshwater flux changes. These studies show that part of the model spread in $\Delta\zeta$ arises from the spread of surface flux changes predicted by AOGCMs (Bouttes and Gregory, 2014), especially regarding the amplitude of the changes.

However, the FAMOUS experiments tend to be similar in their patterns of change; they do not reproduce the diversity of patterns of $\Delta\zeta$ in the AOGCMs supplying the surface flux perturbations. The unexplained model spread in patterns and amplitude of $\Delta\zeta$ must arise from dependence on the ocean model formulation and unperturbed state. These aspects are so far largely unexplored and need further constraint, but comparisons of the ocean response in AOGCMs are complicated by their different predictions of changes to surface fluxes experienced by the ocean.

The Flux-Anomaly-Forced Model Intercomparison Project (FAFMIP) aims to isolate the ocean uncertainty by comparing results from experiments with AOGCMs of CMIP6, the phase of CMIP which is now beginning (Eyring et al., 2016). In the FAFMIP experiments, model-independent surface flux perturbations are imposed on the ocean. At the time of writing there are ten modelling groups who plan to run FAFMIP experiments as part of their contributions to CMIP6, namely ACCESS (Australia), CCCma/CanESM (Canada), CNRM/CERFACS (France), GFDL (USA), GISS (USA), IPSL (France), MIROC

(Japan), MPI-ESM (Germany), MRI (Japan), UKESM (UK). FAFMIP is an element of the science plan for the World Climate Research Programme (WCRP) Grand Challenge on regional sea-level change and coastal impacts.

The AOGCMs participating in FAFMIP will include new three-dimensional ocean diagnostics of the rates of change of temperature and salinity due to the individual processes which transport heat and salt within the ocean (resolved advection, dianeutral mixing, etc.). Such ocean process-based diagnostics have previously been included in only a small number of models

(*e.g.* Gregory, 2000; Huang et al., 2003; Morrison et al., 2013; Palter et al., 2014; Exarchou et al., 2015; Griffies et al., 2015; Kuhlbrodt et al., 2015; Morrison et al., 2016), and cannot be estimated accurately from other archived data. The FAFMIP experiments and diagnostics will for the first time permit us to attribute differences in the ocean among a wide range of models in the unperturbed state and in $CO_2$-forced climate change to particular processes and aspects of model formulation.

The FAFMIP experiments will provide information on the sensitivity of the AMOC to buoyancy forcing of the magnitude and

pattern of that predicted for $CO_2$ forcing, and will support investigation of the correlation between ocean heat uptake efficiency and the magnitude of the AMOC (Rugenstein et al., 2013; Winton et al., 2014; Kostov et al., 2014). The application of common perturbations to surface fluxes in FAFMIP will provide information about the ocean's role in determining patterns of sea surface temperature change worldwide (of relevance to the Grand Challenge on clouds, circulation and climate sensitivity). Similarly





the results will be of relevance to studies of subsurface ocean temperature change in the vicinity of Greenland and Antarctic
ice-shelves (Yin et al., 2011; Spence et al., 2014; Stewart and Thompson, 2015), where warming may promote basal melting
of ice-shelves and consequent sea-level rise through the effect on ice-sheet dynamics (of relevance to the Grand Challenge on
melting ice and global consequences, as well as sea-level).

FAFMIP will thus help with understanding and accounting for the spread in simulated ocean responses in general to changes
in surface fluxes resulting from $CO_2$ forcing. In the next section we describe the design of FAFMIP, and in the following
section we present preliminary results from experiments that have been carried out in a small number of existing AOGCMs to
test the design.

## 2 Design

### 2.1 AOGCMs and surface flux perturbations

The atmosphere and ocean are a tightly coupled system, especially through the interaction of surface heat flux and SST. It
typically requires millennia of "spin-up" integration of an AOGCM with constant atmospheric composition (the pre-industrial
control experiment, denoted "piControl") to reach an approximately steady state in the deep ocean, owing to its large heat
capacity and weak thermal connection to the surface (Danabasoglu, 2004; Stouffer, 2004; Sen Gupta and England, 2004; Banks
et al., 2007; Sen Gupta et al., 2013). Even then, a small "climate drift" may persist. Experiments have been done successfully
in which surface fluxes from one climate state of an AOGCM are transplanted into a simulation of another climate state of the
same AOGCM (Mikolajewicz and Voss, 2000; Gregory et al., 2005). However, if one replaces AOGCM ocean surface fluxes
with real-world estimates or with fluxes diagnosed from *another* model, a large climate drift will result, because they will not
be consistent with the AOGCM's own surface climate. Instead of this, the FAFMIP experiments impose *perturbations*, added
to the surface fluxes that are computed within the AOGCM from the state of the system (Lowe and Gregory, 2006; Bouttes
and Gregory, 2014), technically like the flux adjustment that was formerly used in AOGCMs (Sausen et al., 1988) but with a
different purpose.

The principle of the FAFMIP experiments is that the ocean should respond as it does during an AOGCM climate-change
experiment, as nearly as possible, including interactively simulated atmosphere–ocean feedbacks and unforced variability. The
FAFMIP design contrasts with that of studies using (uncoupled) ocean GCMs, such as the CORE project (*e.g.* Griffies et al.,
2014), in which bulk formulae are used to compute fluxes from prescribed observationally derived surface climate variables,
and the experiments of Marshall et al. (2015), with a prescribed geographically uniform surface heat flux perturbation and
feedback parameter. Those approaches yield valuable and complementary information about the response of the ocean to
perturbations, but are less like the AOGCM projections whose uncertainty we aim to investigate.





**Figure 2.** Annual-mean FAFMIP surface flux perturbations of (a) momentum, (b) heat, (c) water; (d) shows the model-mean change in the surface heat flux $Q$ into the sea-water in the time-mean of the final decade of the faf-heat experiment relative to the control, not including the imposed heat flux perturbation $F$. The ocean area-average of (b) is $1.86\,\mathrm{W\,m^{-2}}$, of (c) $0.072 \times 10^{-6}\,\mathrm{kg\,m^{-2}\,s^{-1}}$ and of (d) $-0.09\,\mathrm{W\,m^{-2}}$. The grey box in (b) is the North Atlantic region to which we refer in Sect. 3.1.

## 2.2 Deriving the surface flux perturbations

Climate-change projection is concerned mostly with scenarios of radiative forcing increasing on decadal timescales. The ide-
135 alised scenario called "1pctCO2" in CMIP6 (and CMIP5), beginning from a piControl state and with atmospheric $CO_2$ con-
centration increasing at $1\%\,\mathrm{yr^{-1}}$, is commonly taken to be indicative of anthropogenic climate change expected during this
century. It is a useful benchmark because it has been studied since the first AOGCM experiments in the early 1990s, while
the more policy-specific scenarios, involving emissions of many species and complicated time-profiles of forcing, have been
revised several times. The transient climate response (TCR) is likewise used for convenient comparison of the magnitude of





climate change, and is defined (Cubasch et al., 2001) as the difference from piControl of the time-mean global-mean surface
air temperature during years 61–80 of 1pctCO2, centred around the time (70 years) at which $CO_2$ reaches double its piControl
concentration.

For consistency with this conventional choice, we obtain the surface flux perturbations for FAFMIP from years 61–80 of
CMIP5 1pctCO2 experiments. In experiments with time-dependent forcing scenarios, the geographical pattern of sea-level
change is fairly constant in time, but has increasing amplitude (Perrette et al., 2013; Bilbao et al., 2015), as is often assumed
for surface air temperature and other surface quantities (Santer et al., 1990; Huntingford et al., 2000; Collins et al., 2013). To
investigate the causes of the patterns we therefore do not need to include interannual variation in the surface flux perturbations.
In the FAFMIP experiments, the perturbations are imposed from the start and held constant (apart from their seasonal cycle).
Tests with the FAMOUS AOGCM indicated that similar geographical patterns of sea-level change result from time-dependent
flux perturbations from 1pctCO2 experiments as from the time-independent FAFMIP flux perturbations.

The surface flux perturbations are derived from a set of 13 CMIP5 AOGCMs for which all the required diagnostics are
available, namely CNRM-CM5, CSIRO-Mk3-6-0, CanESM2, GFDL-ESM2G, HadGEM2-ES, MIROC-ESM, MIROC5, MPI-
ESM-LR, MPI-ESM-MR, MPI-ESM-P, MRI-CGCM3, NorESM1-ME, NorESM1-M. More AOGCMs could have been in-
cluded for some types of perturbation, but it was decided to use this restricted but consistent set of AOGCMs for all per-
turbations, in order to permit comparison with the model-mean change in sea-level and other quantities from the same set of
AOGCMs. The diversity in the surface flux changes from the individual CMIP5 AOGCMs is illustrated by Bouttes and Gregory
(2014) (their Fig. 2). For each of four types of surface flux (zonal and meridional momentum, heat and freshwater, Sect. 2.3), a
difference field for each model is computed between the climatological monthly time-means of years 61–80 of 1pctCO2, using
the first member in cases of an ensemble, and of the corresponding 20 years of piControl, then interpolated to a common 1°
latitude–longitude grid. Finally, the mean of the models is calculated.

The resulting model-mean fields for use in the FAFMIP experiments are stored in CF-netCDF files at `www.met.reading.`
`ac.uk/∼jonathan/FAFMIP`. They are monthly means, which can be regarded as applying at the middle of the month, and
it is recommended to interpolate linearly between them in time to obtain updates at the atmosphere–ocean coupling interval.
Horizontal interpolation to the required ocean model grid may not exactly preserve the global integral but the differences are
not likely to be important.

### 2.3 Experiments

The FAFMIP experiments (Table 1) branch from piControl and have piControl boundary conditions (atmospheric composition,
solar irradiance, land surface, etc.). The best point to branch would be the same point as the 1pctCO2 experiment, with which
FAFMIP results may be compared. The experiments are proposed as 70 years long, but because a large perturbation is switched
on instantaneously at the start, useful results could be obtained from shorter integrations of computationally expensive models.
During the first several decades of a typical AOGCM 1pctCO2 integration, the global-mean surface air temperature and net
heat flux into the ocean rise roughly linearly in time (*e.g.* Gregory and Mitchell, 1997). The flux perturbations in FAFMIP





integrations are typical of year 70 of a 1pctCO2 experiment. Therefore 70 years of a FAFMIP integration will apply roughly
the same time-integral forcing to the ocean as $70\sqrt{2} \simeq 100$ years of a 1pctCO2 integration.

Three experiments are required for participation in FAFMIP (tier 1):

In **faf-stress** we impose a perturbation in surface zonal and meridional momentum flux *i.e.* windstress (Fig. 2a), created
from the CMIP5 diagnostics of surface downward fluxes of eastward (`tauu`) and northward (`tauv`) momentum. Its dominant
feature is the increase in westerly windstress in the Southern Ocean. The stress perturbation is added to the momentum balance
of the ocean water surface. It should not directly perturb any turbulent mixing scheme that depends on the windstress, and
should not be applied to the sea-ice momentum balance, although presumably the sea-ice velocity will be indirectly affected.

In **faf-heat** we impose a perturbation on the heat flux into the sea-water surface (Fig. 2b), created from the CMIP5 diagnostic
of surface downward heat flux in sea-water (`hfds`) *i.e.* the sum of net downward radiative fluxes, sensible and latent heat fluxes
to the atmosphere, and heat fluxes between sea-ice and sea-water. The heat flux perturbation is strongly positive in the North
Atlantic and in the Southern Ocean. Imposing a heat flux perturbation in an AOGCM by adding it to the ocean surface layer
alters the SST and thus modifies the surface heat flux so as to oppose the perturbation. Such a strong negative feedback
does not occur with the momentum flux, which is only fairly weakly affected by the sea-water surface velocity, nor with the
freshwater flux, which does not depend on the surface salinity. The method for implementing faf-heat is the one used by Bouttes
et al. (2014), described below and compared with alternatives (Sect. 2.4); it is intended to avoid this negative feedback, but
permits feedbacks due to ocean circulation change. The method allows us to partition ocean temperature change between the
effects of local addition of heat and changing heat transports, using three-dimensional ocean tracer fields of "added heat" and
"redistributed heat" (Sect. 2.5).

In **faf-water** we impose a perturbation on the freshwater flux into the sea-water surface (Fig. 2c), created from the CMIP5
diagnostic of water flux into sea water (`wfo`) *i.e.* the sum of precipitation, evaporation, river inflow and water fluxes between
floating ice (sea-ice and icebergs) and sea-water. Its pattern is dominated by that of precipitation change, being positive near
the Equator and at mid- to high-latitudes, and negative in the subtropics. In the Arctic there is also increased water input from
river inflow, and a pronounced band of reduced water input from melting along the sea-ice margin, which retreats to higher
latitude in the $2 \times CO_2$ climate.

Two further experiments are recommended (tier 2):

In **faf-all** the surface flux perturbations of momentum, heat and freshwater are simultaneously applied, using the same
method for heat as in the faf-heat experiment.

In **faf-passiveheat** a surface flux equal to the surface heat flux perturbation of the faf-heat experiment is applied instead to
a passive "added heat" tracer (Sect. 2.4), initialised to zero. This tracer does not affect the model evolution, so the experiment
is equivalent to piControl, with an extra diagnostic tracer. Comparison of faf-passiveheat with faf-heat will allow the effect on
the distribution of the added heat from changes in ocean heat transport to be assessed, because these changes do not occur in
faf-passiveheat.

Apart from the partial suppression of changes in surface heat flux in faf-heat (discussed above and in the next section), the
surface fluxes of momentum, heat and freshwater are computed as usual in the AOGCM. In general they will all differ from





the piControl state because of climate change caused by applying the perturbation fluxes to the ocean. In models where the sensible heat content of ocean surface water fluxes (precipitation, evaporation and runoff) is considered, faf-water will in effect

also impose a small heat flux perturbation. Further technical notes on the implementation of each of the experiments can be found at www.met.reading.ac.uk/~jonathan/FAFMIP.

### 2.4 Treatment of the surface heat flux

In this section we consider methods for treating the surface heat flux in faf-heat and faf-all. The methods differ regarding the calculation of the net surface heat flux $Q$ from the atmosphere and sea-ice into the ocean water computed by the AOGCM from

its prognostic state. We refer to the method used by Bouttes et al. (2014) as "B", and compare it with two alternatives, referred to as "A" and "C", with experiments using the HadCM3 AOGCM (Table 2).

In all methods, the net surface heat flux applied to the top-layer $\theta$ in faf-heat is $Q + F$, where $F$ is the FAFMIP prescribed heat flux perturbation, and $\theta$ stands for the ocean model temperature field, either potential or conservative, whichever is used in the equation of state to compute density. Let us write $Q = Q_c$ for the piControl experiment and $Q = Q_p$ for faf-heat. In both

experiments there is unforced interannual variation in $Q$, while the prescribed $F$ has no interannual variation (although it does have a seasonal cycle). The climatological mean difference in net surface heat flux between the experiments is

$$Q_+ = \langle Q_p \rangle + F - \langle Q_c \rangle = \langle \Delta Q \rangle + F, \tag{3}$$

where $\Delta Q = Q_p - Q_c$ and $\langle \rangle$ indicates a climatological time-mean. The aim is that $Q_+$, the difference in surface heat flux between faf-heat and piControl, should equal $F$, the CMIP5 model-mean difference in surface heat flux between the $2 \times CO_2$

climate (in 1pctCO2) and piControl.

In **method A**, the heat flux perturbation $F$ is added to the top layer in the prognostic equation for $\theta$, and the heat fluxes between atmosphere, sea-ice and ocean are calculated as usual in the AOGCM. Since $F > 0$ in large regions and in the global mean (Fig. 2b), surface air temperature generally rises, causing a negative change $\Delta Q$ in the net surface heat flux into the ocean. This change opposes $F$, so ocean area-mean $\overline{Q_+} < \overline{F}$ (Eq. 3). In the HadCM3 experiment with method A, global-mean

surface air temperature rises by 0.8 K (Fig. 3b), and the ocean area-mean $\langle \overline{\Delta Q} \rangle = -0.81$ W m$^{-2}$, while $\overline{F} = 1.86$ W m$^{-2}$. Thus only $1 + \langle \overline{\Delta Q} \rangle / \overline{F} = 56\%$ of the heat flux perturbation is added to the ocean. Locally $\langle \Delta Q \rangle$ is generally of opposite sign to $F$ (compare Fig. 2b and 3a), as expected, and it is of particularly large magnitude in the North Atlantic.

In **method B** (further discussed in section 2.5), we introduce a passive tracer $T_R$ i.e. one which does not affect density. It is initialised to $\theta$ at the start of the experiment, and subsequently transported by all the same processes as $\theta$. The model's surface

heat flux $Q$ is applied to $T_R$ as well as to $\theta$, but $T_R$ does not feel the heat flux perturbation $F$. The critical difference from method A is that the SST for computing $Q$ is supplied by $T_R$ instead of $\theta$, and is therefore not directly affected by $F$. This mitigates the feedback in which $\Delta Q$ opposes $F$. Similarly $T_R$ is used instead of $\theta$ in calculations of the heat fluxes between the ocean and sea-ice, so that $F$ does not directly affect the sea-ice heat budget. If $F = 0$, $T_R$ evolves like $\theta$ and the climate will be the same as in piControl. Method B is more complicated than method A because of the need for $T_R$ and small modifications to



**Figure 3.** (a,c,e) The change $\Delta Q$ (W m$^{-2}$) in the surface heat flux into sea-water in the time-mean of the 70 years of the HadCM3 faf-heat experiment relative to the control, not including the imposed heat flux perturbation, in methods A, B and C; (b,d,f) Annual timeseries of change relative to control in global-mean surface air temperature, ocean volume-mean temperature, and maximum of AMOC streamfunction, with the three methods.





the coupling to atmosphere and sea-ice submodels. However, extra tracers are a standard mechanism in many OGCMs because of the role in ocean biogeochemistry and for diagnostics such as idealised age and chemical species.

     In the HadCM3 experiment with method B, the change in global-mean surface air temperature is prevented (Fig. 3b), and ocean area-mean $\langle \overline{\Delta Q} \rangle = -0.21 \, \mathrm{W \, m^{-2}}$, so 90% of the heat flux perturbation is added to the ocean, causing a greater increase in ocean heat content than in method A (Fig. 3d). Locally $\Delta Q$ is no longer markedly anticorrelated with $F$ (compare Fig. 2b

and 3c). Whereas method A puts less heat than the intended $F$ into the North Atlantic, method B puts more than intended, as a result of changes in ocean circulation. The weakening of the AMOC reduces advective heat convergence to the North Atlantic. In the unmodified AOGCM, as in the real world, this causes a cooling tendency to regional SST, and the surface heat flux into the ocean will tend to increase in consequence. Winton et al. (2013) show that about one-third of the reduction in heat convergence may thus be offset by an increase in surface heat flux. This mechanism is presumably at work in the

CMIP5 1pctCO2 experiments from which the faf-heat $F$ has been calculated, and $F$ therefore includes an enhancement due to reduction of advective heat convergence in those models. But the mechanism operates in faf-heat as well, because weakening of the AMOC will reduce the convergence of $T_R$, from which $Q$ is calculated. Hence this phenomenon is exaggerated in method B, making $Q_+$ larger than intended. The change in advection means also that $Q_+$ does not have the intended geographical distribution.

In **method C** for faf-heat, the AOGCM uses climatological monthly time-means of SST and sea-ice from piControl, instead of the prognostic state of the system, to compute the ocean surface heat flux $Q$. The sea-surface conditions evolve in response to $F$ in the ocean submodel, but these changes do not affect the atmosphere submodel. Because this method suppresses the interaction between surface climate and atmosphere, an ocean climate drift results even if $F = 0$.

     In our HadCM3 test of method C, the surface climate for $F = 0$ stabilises within about 100 years. The ocean area-mean

SST is 1.4 K warmer than in the HadCM3 control, with cooling of more than 2 K in the North Atlantic, although the AMOC is unaffected in strength, and warming of more than 2 K in low latitudes. The sea-surface conditions applying to the ocean and atmosphere are therefore markedly different, since the latter is prescribed unchanged from the control. In the global mean the ocean warming penetrates to about 500 m depth, with both cooling and warming of more than 1 K in magnitude at greater depths in high northern latitudes. The ocean area-mean surface salinity increases by about 0.5 PSU. These changes are

comparable in magnitude with those that result from $2 \times CO_2$ forcing, meaning that for method C, unlike method B, a new control experiment with $F = 0$ is required in parallel to faf-heat to evaluate the response to the perturbative $F$.

     In method C, the effect of $F$ on $\Delta Q$ via SST is eliminated, and $Q_+$ is close to $F$. In HadCM3 with method C 97% of the global-mean $F$ is added to the ocean, whose heat content increases slightly more than in method B (Fig. 3d). However, $\Delta Q$ is not zero everywhere (Fig. 3e), because the faf-heat and corresponding piControl integrations have different unforced variability

in the atmosphere, and because changes in ocean surface velocity are seen by the atmosphere.

     Since method C applies the heat flux perturbation accurately to the North Atlantic, without allowing the strong local feedback on $Q$, its simulation of the AMOC decline may give the best estimate of the response to the intended $F$. Compared with method C, the AMOC weakening is too small in method A and too large in method B (Fig. 3f). We note that when Bouttes et al. (2014, their Fig. 5), applied surface heat flux perturbations from CMIP5 AOGCMs to the FAMOUS AOGCM using





method B, the weakening of the AMOC in FAMOUS was not systematically stronger than in the CMIP5 AOGCMs. This indicates that strength of advection feedback on $Q$ is model-dependent, as we also see later (Sect. 3.1).

Although method C is arguably most accurate, it has the disadvantages that it is more computationally expensive, because of the need for a new control integration, the ocean climate state is different from the unmodified AOGCM whose response we wish to investigate, and the physical interaction between atmosphere and ocean is unrealistically suppressed, including feedbacks which could be of interest. We therefore adopt method B for faf-heat.

## 2.5 Added and redistributed heat

Changes in $\theta$ in method B can be partitioned into those due to modified tracer transport processes (due to change in circulation, diffusion, etc.) and those due to added heat (following Banks and Gregory, 2006; Xie and Vallis, 2012). We are interested in the evolution of the climatological state, so all terms should be interpreted as climatological time-means (and we omit $\langle \rangle$ for the sake of legibility). As in the previous section, we use subscripts $c$ and $p$ to denote variables in the piControl and perturbed (faf-heat) experiments respectively. By $\Phi(\theta)$ we denote the net heat convergence due to *all* heat fluxes in the interior of the ocean, both resolved and parametrised subgridscale. The function $\Phi$ depends on diffusivities and other attributes of the model state which affect heat transport, as well as the velocity.

In the piControl experiment,

$$\frac{\partial \theta_c}{\partial t} = Q_c + \Phi_c(\theta_c), \tag{4}$$

setting the volumetric heat capacity to unity for convenience. The $\theta$ field is three-dimensional but $Q$ applies only at the surface.

In faf-heat $\theta = \theta_p$ is affected by the imposed heat flux perturbation $F$ as well as by the atmosphere–ocean heat flux $Q = Q_p$ simulated by the AOGCM, so

$$\frac{\partial \theta_p}{\partial t} = Q_p + F + \Phi_p(\theta_p), \tag{5}$$

where $\Phi_p$ is a different function from $\Phi_c$ because of changed velocities, diffusivities, etc.

The redistributed heat tracer $T_R$, which we described for method B in section 2.4, is initialised to $\theta_c$ and has $Q_p = Q_c + \Delta Q$ as its surface flux, so its evolution equation is

$$\frac{\partial T_R}{\partial t} = Q_p + \Phi_p(T_R). \tag{6}$$

Since $T_R$ is initialised to $\theta_c$, we write $T_R = \theta_c + \Delta T_R$ *i.e.* $\Delta T_R = 0$ initially. Let us also split $\Phi_p$ into $\Phi_c + \Delta \Phi$. (For example, this splits the advective heat convergence into the part $-\nabla \cdot (T_R \mathbf{v}_c)$ due to the piControl velocity field $\mathbf{v}_c$ and the part $-\nabla \cdot (T_R \Delta \mathbf{v})$ due to the change in the velocity field with respect to the piControl.) These decompositions assume that the heat convergence function depends linearly on the relevant variables of the climate state and acts linearly on the tracers. In that case

$$\frac{\partial T_R}{\partial t} = Q_c + \Phi_c(\theta_c) + \Delta Q + \Delta \Phi(\theta_c) + \Phi_p(\Delta T_R) = \Delta Q + \Delta \Phi(\theta_c) + \Phi_p(\Delta T_R) \tag{7}$$





if the piControl is a steady state, so that $Q_c + \Phi_c(\theta_c) = 0$ (Eq. 4). $T_R$ is called the "redistributed heat" tracer by Xie and Vallis (2012), because it diagnoses the effect of changes in tracer transport processes (changes in circulation, diffusivities, etc., giving rise to $\Delta\Phi$) on the unperturbed $\theta_c$. If $\Delta\Phi$ vanishes, and assuming $\Delta Q = 0$ as well, $\Delta T_R = 0$ always, meaning that $T_R$ evolves identically to $\theta_c$ and thus they remain equal. Changes in ocean heat transport may induce a non-zero $\Delta Q$ (Sect. 2.4), which will affect $T_R$ as well.

The "added heat" tracer $T_A$ in faf-heat is initialised to zero (so $\Delta T_A \equiv T_A$) and has $F$ as its surface flux (we note that heat is added in the global mean, although $F$ is not positive everywhere, as seen in Fig. 2b). It shows where the heat flux applied as a perturbation is stored in the ocean. Its evolution equation is

$$\frac{\partial T_A}{\partial t} = F + \Phi_p(T_A). \tag{8}$$

so $T_A = 0$ always if $F = 0$. This tracer is similar to the "passive anomalous temperature" of Banks and Gregory (2006), whose
experimental design was different. The added heat tracer is included also in the faf-passiveheat experiment, where its surface source is the same but its evolution is different, because it is subject to the same circulation and subgridscale processes as in the control state, and $\Phi_c$ replaces $\Phi_p$ in equation 8.

Considering Eqs. 5, 6 and 8, we see that

$$\frac{\partial \theta_p}{\partial t} = \frac{\partial T_R}{\partial t} + \frac{\partial T_A}{\partial t}. \tag{9}$$

Thus we can interpret changes in ocean heat content in faf-heat as the sum of redistribution (including the effect of $\Delta Q$) and addition. In practice to achieve exact equality may not be possible due to non-linearities in the implementation of tracer transport operators.

Careful formulation is required to ensure that $Q$ is applied in the same way to $\theta$ and $T_R$, and some differences may be unavoidable, depending on model formulation. In particular, absorption of solar radiation should occur with the same vertical
profile for both (assuming that some of it penetrates the top layer), and the same heat flux should be applied to both of them for evaporation and precipitation (if the sensible heat content of these water fluxes is considered in the model). If the same amount of heat is extracted from both tracers for frazil sea-ice formation, $\theta$ may sometimes fall below freezing point, requiring special treatment of the equation of state; on the other hand if $\theta$ and $T_R$ are separately kept above freezing, there will be a difference in the heat fluxes implied. Further technical notes can be found at www.met.reading.ac.uk/~jonathan/FAFMIP. It
may be useful to check the implementation of $T_R$ in the model with an experiment in which $F = 0$, which should reproduce the piControl experiment.

## 2.6 Diagnostics

FAFMIP experiments should include standard CMIP6 monthly mean and other diagnostics of atmosphere, ocean and cryosphere, as in the DECK; these provide a large amount of information which will support many kinds of analysis that cannot
be anticipated in detail. The standard ocean diagnostics are described in detail for the Ocean Model Intercomparison Project (OMIP) by Griffies et al. (submitted) in this issue, and in Table 3 we list a subset of particular importance to FAFMIP, for which they are priority 1 as monthly means. We refer to them here by their CMIP "short names".





Analysis of sea-level change and ocean heat uptake will use diagnostics of sea-level, ocean temperature and salinity (`zos`, `zostoga`, `thetao` or `bigthetao`, `thetaoga` or `bigthetaoga`, `opottempmint` or `ocontempmint`, `so` and `somint`, where the choice of alternatives depends on whether the prognostic ocean temperature is potential or conservative). Analyses of the AMOC will use the overturning streamfunction (`msftmyz` or `msftyyz`). The faf-heat and faf-passiveheat experiments should include monthly means of the added heat tracer $T_A$ (`pathetao` or `pabigthetao`), and faf-heat should include monthly means of the redistributed heat tracer $T_R$ (`prthetao` or `prbigthetao`).

Analysis of ocean tracer budgets will use the ocean surface heat and water fluxes requested as standard CMIP monthly diagnostics. Surface fluxes affect only the top layer of the ocean, except for shortwave (solar) radiation, which penetrates more deeply (diagnosed by `rsdoabsorb`). The net surface heat and water fluxes into sea water (`hfds` and `wfo`) are particularly useful, because model-dependent details of implementation, especially regarding sea-ice, can make it an intricate or impossible task to compute the net fluxes from other CMIP diagnostics. The net surface flux diagnostics should be as computed by the model, not including the imposed flux perturbations *i.e.* `hfds` should omit the FAFMIP heat flux perturbation and `wfo` should omit the FAFMIP water flux perturbation.

Intercomparative analysis of ocean interior change is a priority for FAFMIP, motivating the introduction of the three-dimensional process-based tendency diagnostics for prognostic temperature and salinity (Table 4). These diagnostics are described in detail in Sect. 9 of Griffies et al. (submitted). Different models parametrise interior transports in many ways, so for the purpose of intercomparison it is necessary to aggregate them into broad classes. We distinguish advection by the model velocity field, parametrised eddy advection (mesoscale and submesocale if treated separately), mesoscale diffusion (by eddies along neutral or isopycnal surfaces), and dianeutral mixing (including diapyncal diffusion, convection and boundary-layer mixing). In addition there is a net tendency diagnostic, whose time-integral over any period should equal the change in the prognostic between the start and end of that period. The difference between the net tendency and the sum of the individual process diagnostics will yield a residual that accounts for any other schemes not separately identified, including the effect of surface fluxes.

The tendency diagnostics are expressed as rates of change of heat and salt content in gridcells *i.e.* $\partial(mC_p\theta)/\partial t$ and $\partial(mS)/\partial t$, where $S$ is salinity, $m$ is the mass per unit area of the gridcell and $C_p$ the specific heat capacity. In Boussinesq models with fixed cell thicknesses, $m$ is a constant for each gridcell, but otherwise it is variable. The tendency diagnostics are requested at priority 1 as annual means, and at priority 2 as monthly means for analysis of high-frequency variability, recognising that this implies a substantial amount of storage. As well as in the FAFMIP experiments, these diagnostics should be included in the DECK 1pctCO2 and abrupt4xCO2 experiments, and in the piControl experiment, at least in the portion which is parallel to the FAFMIP experiments. This will give information which has never previously been available for AOGCMs in general, concerning the roles of the various interior processes in the maintenance of the steady state, unforced variability, and the response to climate change.



**Figure 4.** Annual timeseries: top row, ocean volume-mean temperature change (K) with respect to the corresponding year of the control; middle row, maximum of the Atlantic meridional overturning streamfunction (Sv), with the 70-year time-mean of the control experiment shown as a dotted line; bottom row: $\sigma_\zeta$ (m), the spatial standard deviation of $\Delta\zeta$, the dynamical sea-level change relative to the 70-year time-mean in the control experiment, with the time-mean $\sigma_\zeta$ for the control run shown as a dotted line.



## 3 Preliminary results

To test the design, the FAFMIP experiments have been carried out by five groups using existing models from previous phases of CMIP (Table 2). These preliminary experiments did not include the process-based tendency diagnostics described in Sect. 2.6. In order to demonstrate the usefulness of the experiments and stimulate interest in analysis, we present an overview of the results in this section.

### 3.1 Time-dependence

Since the imposed FAFMIP surface flux perturbations have no interannual trend or variability, we expect that the ocean will gradually evolve towards a new steady state, as its three-dimensional density and velocity fields adapt to balance the modified surface boundary conditions. The surface fluxes will also evolve as part of this process, because they depend on the surface climate. Timeseries of global-mean quantities give a useful indication of the approach to the steady state.

The global-mean surface air temperature change $\Delta T$ with respect to control reaches a steady state in about 30 years in all the FAFMIP experiments, with time-means within $\pm 0.3$ K in most cases (not shown). These are small changes compared with that expected in response to 1pctCO2, in which $\Delta T$ after 70 years (at the time of $2 \times CO_2$), referred to as the "transient climate response", has a range of 1.0–2.5 K for CMIP5 AOGCMs. Note that the heat flux perturbation of faf-heat and faf-all does not affect $\Delta T$ directly, because $T_R$ is used to supply the SST for the surface climate (Sect. 2.4). Despite the small global-mean $\Delta T$, substantial regional changes develop in surface air temperature in all the experiments, and in faf-water all models show a widespread surface cooling. We discuss these points below (Sect. 3.2) when considering changes in ocean interior temperature.

The imposed surface heat flux perturbation in faf-heat is unopposed by increased heat loss to space, because global-mean surface air temperature change is suppressed in method B. Consequently ocean volume-mean temperature rises continuously during faf-heat (top row of Fig. 4). An ocean volume-mean temperature change of 0.1 K is equivalent to an increase in ocean heat content (OHC) of 0.53 YJ, a time-mean heat input of 0.66 W m$^{-2}$ averaged over the ocean surface for 70 years, and would produce GMSLR due to thermal expansion of 64 mm (using the CMIP5 model-mean expansion efficiency of heat). By comparison, the change in ocean volume-mean temperature is very small in faf-stress and faf-water. In these experiments, the global OHC is redistributed, as discussed below, with hardly any net change.

The timeseries of change in the AMOC (middle row of Fig. 4) are of interest because of its importance to sea-level change in the North Atlantic and regional climate change in Europe. The faf-stress and faf-water experiments show that the perturbations to surface momentum and water fluxes typical of $CO_2$-induced climate change have little influence on the AMOC. A dominant influence of heat flux change on the AMOC in response to $CO_2$, which can be seen in faf-heat, has been inferred in some earlier investigations (Rahmstorf and Ganapolski, 1999; Mikolajewicz and Voss, 2000; Gregory et al., 2005), which did not include experiments with heat flux perturbations, although water fluxes dominated in other models (*e.g.* Dixon et al., 1999). Heat flux perturbations have also been found to be the dominant influence on AMOC variability (Delworth et al., 1993; Griffies and Tziperman, 1995; Delworth and Greatbatch, 2000).



As we discuss above (Sect. 2.4), the faf-heat design exaggerates the increase in the surface heat flux in the North Atlantic compared with 1pctCO2. The mean of $F$ over ocean area in the North Atlantic within 80°W–10°E and 30–65°N is 0.57 W m$^{-2}$. The mean of $\Delta Q$ in this region is 0.48 W m$^{-2}$ in the model mean (shown for each model in Table 2), so on average the feedback

nearly doubles the heat input to this region. The imposed $F$ and the feedback $\Delta Q$ have remarkably similar distributions (Fig. 2b,d). In the rest of the world, the model-mean $\Delta Q$ is relatively small. Its global mean of $-0.09$ W m$^{-2}$ is much smaller than the global-mean $F$ of 1.86 W m$^{-2}$ (shown for individual models in Table 2).

Both the magnitude and the time-profile of the AMOC weakening in faf-heat are model-dependent (Fig. 4). We presume that the feedback on the heat input also exaggerates the weakening of the AMOC in faf-heat, which is larger than at the time of

$2 \times CO_2$ (using the time-mean of years 61–80) in 1pctCO2 experiments with the same AOGCMs (Table 2, $\Delta$AMOC columns). Another reason for a larger response than in 1pctCO2 is that the heat flux perturbation, which is consistent with $2 \times CO_2$, is applied from the start of the faf-heat experiment.

Although $\Delta Q$ always increases the heat flux added to the North Atlantic, the model spread in $\Delta Q$ is relatively small (Table 2), so the net addition of heat $F_+ = F + \Delta Q$ in the North Atlantic in faf-heat is quite similar in the four models.

Moreover, although the AMOC weakening always is larger in faf-heat than in 1pctCO2, it correlates between faf-heat and 1pctCO2 across the four AOGCMs, and they are in the same rank order. These points suggest that the faf-heat results may be used to investigate the spread of AMOC weakening in $CO_2$-forced experiments, despite the amplification.

We note that the water flux perturbation does not include freshwater input arising from loss of mass by the Greenland ice sheet, because this effect is mostly not included in the CMIP5 AOGCMs from which it was derived. Several studies have

evaluated the AMOC response to a freshwater flux of ∼0.1 Sv into the ocean in the vicinity of Greenland. They report a range of results for AMOC weakening, for example by about 2 Sv after several centuries (Vizcaíno et al., 2010), by $1.1 \pm 0.6$ Sv by the end of the 21st century in a comparison of five models (Swingedouw et al., 2015), and by about 5 Sv in fifty years in a comparison of models with 1° and 0.1° resolution (Weijer et al., 2012). In the last study, with the eddy-resolving (0.1°) resolution, the AMOC weakening was about 10 Sv when the water flux was applied uniformly over the Atlantic within 50–

70°N, following the design of an earlier model intercomparison (Stouffer et al., 2006), in which the AMOC weakening after 100 years showed a large model spread of 0–10 Sv. An addition of 0.1 Sv is a very large perturbation in comparison with the rate of mass loss from the ice-sheet during 2002–2011, which was about 200 Gt yr$^{-1}$ (Vaughan et al., 2013), equivalent to 0.6 mm yr$^{-1}$ of GMSLR, and 0.006 Sv of freshwater added to the ocean. The area-integral of the FAFMIP water flux perturbation field over 50–70°N and 70°W–30°E in the Atlantic is 0.007 Sv.

To monitor the change in regional sea-level, we compute the timeseries of area-weighted spatial standard deviation $\sigma_\zeta(t)$ of annual-mean $\Delta\zeta$ (bottom row of Fig. 4). This quantity is also the spatial standard deviation of $\Delta\eta(\mathbf{x},t)$, since $\Delta\eta$ and $\Delta\zeta$ differ only in their global means. Because of unforced variability within the climate system, local sea-level in any given year will differ from its long-term mean, so the control time-mean of $\sigma_\zeta$ is not zero. It is model-dependent and in the range 0.02–0.06 m (in agreement with Bilbao et al., 2015, their Fig. 2). In the perturbed FAFMIP experiments, a forced pattern of $\Delta\zeta$ gradually

emerges in addition to and independent of the unforced interannual variability, and $\sigma_\zeta$ thus rises above its control value. In faf-stress and faf-water it levels off within about 30 years, showing an increase of ∼0.01 m, which indicates that sea-level change



is not pronounced or widespread. In faf-heat the increase takes longer and is larger; after 70 years it has reached 0.06–0.10 m and has not stabilised. This means that the pattern of $\Delta\zeta$ is increasing in amplitude.

## 3.2 Spatial patterns

To describe the eventual response to the surface flux perturbations, we consider the state reached by the end of the experiments, as shown by the difference between the time-mean of the last decade, years 61–70 and the corresponding decade of the control experiment. As was intended by the experimental design, the FAFMIP results exhibit the same major features of dynamic sea-level change as found in previous studies for 1pctCO2 and other scenarios (Fig. 5). The heat flux perturbation produces the largest local changes in $\zeta$. It is interesting to note, by contrast, that over the last couple of decades (the period of continuous

satellite sea-level altimetry) the largest regional trends in sea-level are caused by momentum flux changes (windstress) in the Pacific (England et al., 2014; Griffies et al., 2014). The east–west contrast in the Pacific is not a pattern predicted in response to $CO_2$ forcing by AOGCMs (Bilbao et al., 2015).

The increased sea-level gradient across the ACC (positive $\Delta\zeta$ to the north and negative to the south) has contributions from both momentum and heat, and is somewhat counteracted by water (Fig. 5) (Bouttes and Gregory, 2014; Saenko et al., 2015).

There is warming locally of up to ∼1 K in surface air temperature near Antarctica in both faf-stress and faf-heat, and local cooling in faf-water (Fig. 6). From the similarity of $\Delta\zeta$ and changes in local OHC (the vertical integral of $\theta$ expressed as heat) in the Southern Ocean in faf-stress (Figs. 5a and 7a), we infer that the effect of the momentum flux perturbation on sea-level is predominantly thermosteric rather than halosteric. Although the momentum and heat flux perturbations are the same in all models, the meridional gradient in $\Delta\zeta$ across the ACC is model-dependent (Fig. 5b,d). Subtracting the global-mean OHC

increase from faf-heat reveals that the distribution of OHC change is remarkably similar in faf-stress and faf-heat (Figs. 7a,d).

The water flux perturbation is positive at high latitudes, and causes positive $\Delta\zeta$ in the Arctic and near Antarctica (Figs. 5e,f). In the Arctic there is reduced OHC and that $\Delta\zeta$ is predominantly halosteric *i.e.* due to reduced salinity, caused by increased freshwater input (Figs. 2c and 7c). In the Antarctic $\Delta\zeta$ is partly thermosteric, associated with increased OHC (Fig. 7c). It could arise from suppression of upward convective or diffusive heat loss due to reduction of surface salinity and increased stability of the water column, and causes warming to considerable depth in these latitudes (Fig. 8e). Although the AMOC does not

change substantially, there is an increase in OHC in much of the Atlantic in faf-water (Figs. 7c and 8e), and a reduction north of ∼45°N; this pattern is correlated with (*i.e.* density-compensated by) the change in salinity content.

The water flux perturbation causes surface air temperature to cool over a large fraction of the world (Fig. 6c), by 0.2–0.4 K in the global mean and more than 1 K in some regions. We presume that this is due to the suppression of upward heat transport by

a reduction in surface salinity (Fig. 8f). This leads to a downward redistribution of heat from the surface to layers below a few 100 m (Fig. 8f). A similar tendency to widespread surface cooling was found by Stammer et al. (2011) in response to addition of 0.1 Sv freshwater to the ocean in the vicinity of Greenland. The global integral of the FAFMIP water flux perturbation field is 0.027 Sv and its ocean area-average is very small compared with its local values (Fig. 2c). The cooling also occurs in a modified faf-water experiment with HadCM3 and a water flux perturbation field having zero mean (obtained by uniformly



**Figure 5.** Change in dynamic sea-level $\Delta\zeta$ (m) in the time-mean of the final decade of the FAFMIP experiments relative to the control, model mean on the left, zonal means of individual models on the right. Note that the panels on the right have different scales for the $\Delta\zeta$ axis.



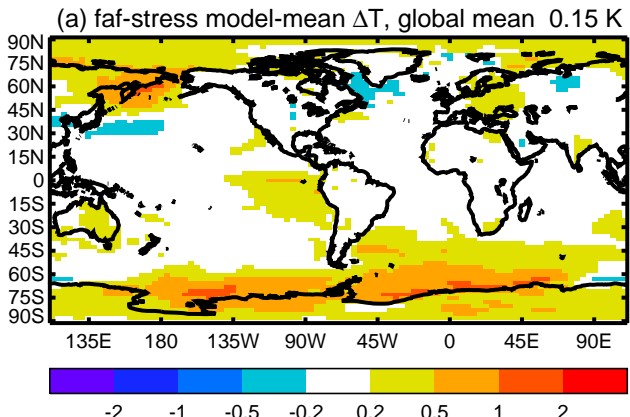

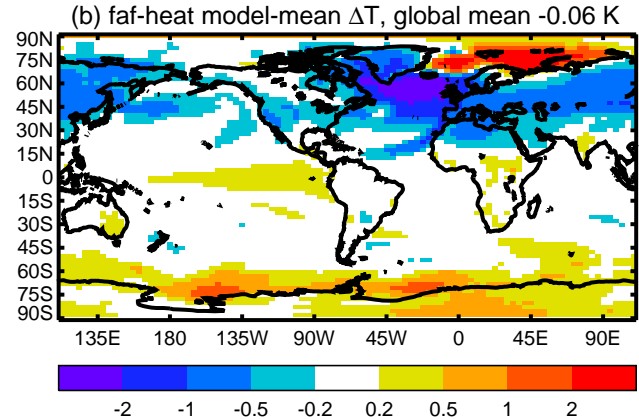

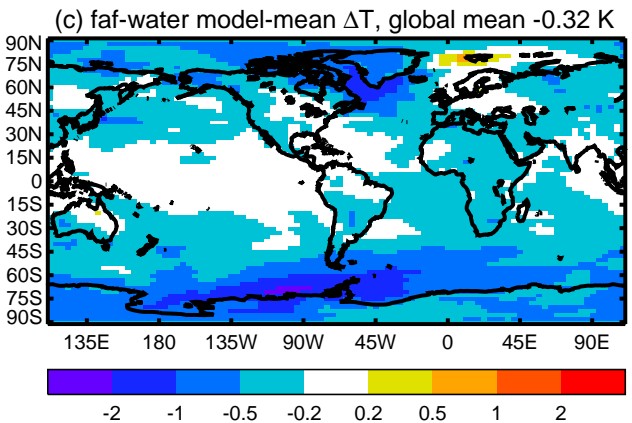

**Figure 6.** Change in surface air temperature (K) in the time-mean of the final decade of the FAFMIP experiments relative to the control.

subtracting the area-average of the standard field), indicating that, at least in this model, the phenomenon is not a response to the global-mean of the perturbation flux, but to its geographical pattern through some non-linear mechanism.

The dipole in $\Delta\zeta$ in the North Atlantic (positive to the north of $\sim$40°N, negative to the south) is mainly due to the heat flux perturbation (Fig. 5). It is consistent with greater increase in OHC to the north of this latitude in the Atlantic (Fig. 7b), and reinforced by changes in salinity (not shown), associated with the weakening of the AMOC in faf-heat (Pardaens et al., 2011;

Stammer et al., 2011; Bouttes et al., 2014; Saenko et al., 2015), which reduces northward salinity advection, thus causing an increase in salinity to the south (negative $\Delta\zeta$) and a decrease to the north (positive $\Delta\zeta$). The reduction of heat transport by the AMOC in faf-heat produces a strong cooling of surface air temperature of more than 2 K locally in the North Atlantic (Fig. 6b), and in similar latitudes of Eurasia and North America, as has been found by many previous studies (*e.g.* Stouffer et al., 2006). The water flux perturbation contributes to the Atlantic dipole as well (Fig. 5d), because it is positive to the north





**Figure 7.** Model-mean change in ocean heat content (GJ m$^{-2}$, the vertical integral of the change in the ocean temperature $\theta$ multiplied by the volumetric heat capacity) in the time-mean of the final decade of the FAFMIP experiments relative to the control. (d) shows the field of (b) with its global mean subtracted.

(reducing salinity, raising $\Delta\zeta$) and negative to the south. The momentum perturbation makes no significant contribution to $\Delta\zeta$ in the North Atlantic.

### 3.3  Addition and redistribution of heat

In faf-heat the added and redistributed heat tracers give us further information about changes in OHC. Since the input is at the surface, the amount of added heat declines with depth in the global mean, with a model-dependent vertical profile (Fig. 9a)

(Kuhlbrodt and Gregory, 2012). The volume-integral varies by $\pm5\%$ across models (1.32–1.46 YJ of added heat, Table 2). Because this is the time-integral of $F$, it should differ among models only because of different land–sea boundaries, ocean area and regridding method.



**Figure 8.** Model-mean change in ocean temperature $\theta$ (K) in the time-mean of the final decade of the FAFMIP experiments relative to the control, model-mean zonal-mean cross-sections on the left, global means as a function of depth of individual models on the right. Note that the panels on the right have different scales for the temperature axis.





**Figure 9.** Tracers of added heat (on the left) and redistributed heat (on the right) in the final decade of the FAFMIP-heat experiment, (a,b) global-mean change in tracer (K) as a function of depth, with different scales for the temperature axis, (c,d) model-mean change in heat content (GJ m$^{-2}$, the vertical integral of the change in tracer multiplied by volumetric heat capacity), (e,f) model-mean zonal-mean cross-sections of the change in tracer (K).





**Figure 10.** Change in ocean heat content ($10^{21}$ J per degree of latitude) relative to the control in the time-mean of the final decade of the faf-heat and faf-passiveheat experiments. The changes in ocean heat content, added heat and redistributed heat are calculated from the integrals over longitude and depth of $\theta$, $T_A$ and $T_R$ respectively, multiplied by volumetric heat capacity. The surface heat flux perturbation $F$ is shown as its integral over longitude and time (for 65 years, to the middle of the final decade), divided by 2 in order to fit on the same axis. The global integrals of $F$ and $T_A$ should be equal.

The greatest surface input of added heat from the heat flux perturbation is to the Southern Ocean (Fig. 2b and 10, grey lines) but the added heat accumulates at lower latitude than the input, due to its wind-driven convergence and subduction centred within 30–45°S (Figs. 10, solid red lines, and 9c,e). Because of this, the OHC increase near Antarctic is relatively small (Fig. 7b), and $\Delta\zeta$ is negative (Fig. 5c,d), while in the southern mid-latitudes, on the north side of the ACC, there is a relatively large addition of heat and positive $\Delta\zeta$.





The vertical profile of temperature change in faf-heat (Fig. 8d), is dominated by the added heat (Fig. 9a). There is a minor influence from redistribution of heat downwards from the surface and upwards from the deep ocean into layers about 500 m

deep (Fig. 9b). The opposite vertical profile of redistribution is seen in faf-stress (Fig. 8b), in which the increased westerly windstress strengthens the overturning circulation and causes heat to converge around 45°S, where it is pumped downward (Fig. 8a). The small vertical gradient in temperature change between 200 and 500 m in faf-heat in GFDL-ESM2M is due to redistribution; it relates to a cooling in the shallow tropics and resembles the response of the same model to volcanic forcing (Stenchikov et al., 2009, their Fig. 3) with the opposite sign.

In faf-heat, heat is redistributed from the mid-latitude gyres, around 30° in both hemispheres, towards the Equator (Fig. 10, blue lines, and 9d,f). The volume-integral of the redistributed heat lies between $-0.12$ and $0.08$ YJ, only a small fraction of the added heat. It is not zero because it is affected by $\Delta Q$.

Marked changes occur in the North Atlantic associated with the AMOC, although they do not dominate the global picture because the Atlantic has a relatively small area. Deep water formation conveys added heat to the deep North Atlantic around

60°N (Fig. 9e). The weakening of the AMOC tends to reduce northward and downward heat transport, causing redistributive cooling throughout the North Atlantic (Fig. 9d,f), except in a narrow band along the east coast of North America, where the weakened northward transport in the boundary current reduces the divergence of heat, increases OHC and enhances $\Delta\zeta$ (Yin et al., 2009; Bouttes et al., 2014). In the deep North Atlantic, negative redistribution outweighs positive addition of heat, and a net cooling results (Fig. 8c).

As intended by construction (Sect. 2.4), the sum of added and redistributed heat is very similar or identical to the change in OHC (Fig. 10, compare black and green lines). The latitudinal distribution of added heat is very similar in faf-heat and faf-passiveheat (Fig. 10, compare red solid and dashed lines), especially in the southern hemisphere. This indicates that the influence of change in transport on the added heat is of second order, as expected. South of 30°S, changes in OHC and added heat are fairly similar in the zonal integral (Fig. 10, compare solid red and black lines) *i.e.* redistribution is relatively small, and

heat uptake is largely passive.

## 4 Summary and plans

The purpose of the flux-anomaly-forced model intercomparison project (FAFMIP) is to analyse the simulated response of the ocean to changes in surface fluxes resulting from $CO_2$ forcing in AOGCMs. The specific interests which motivated the proposal of FAFMIP are

– The magnitude of ocean heat uptake in response to climate change, which determines global-mean sea-level rise due to thermal expansion and influences the transient climate response.

– The geographical patterns of sea-level change due to ocean density and circulation change simulated by the models.

– The weakening of the Atlantic meridional overturning circulation, which affects regional sea-level rise and climate change.



– The ocean's role in determining the patterns of sea surface temperature change, which influences climate sensitivity to $CO_2$.

    – Subsurface warming of the ocean near to the Greenland and Antarctic ice-sheets, where it might enhance basal melting of ice-shelves and hence sea-level rise through the dynamical response of the ice-sheets.

These topics are all aspects of the Earth system response to forcing, and they are of particular relevance to the WCRP Grand
Challenges on regional sea-level rise, melting ice, and climate sensitivity. The motivation for FAFMIP is to find ways of reducing the uncertainty in projections in policy-relevant scenarios, by applying observational constraints and improved physical understanding to refine the models.

In the FAFMIP tier-1 experiments faf-stress, faf-heat and faf-water, prescribed perturbations are applied to the ocean surface in the fluxes of momentum, heat and freshwater respectively. The flux perturbations have a seasonal cycle but no interannual
variation, and are obtained from a model mean of changes simulated in CMIP5 AOGCM experiments at year 70 in 1pctCO2 experiments (with $CO_2$ increasing at $1\% \, \mathrm{yr}^{-1}$). They are thus typical of simulated $CO_2$-forced climate change in magnitude and geographical pattern. The intention of applying the same surface flux perturbations in all AOGCMs in FAFMIP is to reveal the dependence of the response on the ocean model. The FAFMIP tier-1 experiments amount to 210 years of integration, which is a modest requirement compared with many CMIP6 subprojects. There are two tier-2 experiments of 70 years each, one of
which can be achieved by adding a diagnostic to the control experiment, thereby avoiding the need for a separate integration. We have carried out preliminary tier-1 experiments with pre-CMIP6 AOGCMs to test and demonstrate the experimental design. Our models exhibit diversity in the pattern and magnitude of simulated changes, with some common qualitative features.

We find that momentum and water flux perturbation do not affect the AMOC significantly, but the AMOC weakens in faf-heat, by 6–12 Sv depending on model, in response to the heat added to the North Atlantic. The AMOC weakening is reinforced
by a feedback on the surface heat flux whereby, as the AMOC declines, the SST in the North Atlantic tends to cool, so the heat flux from the atmosphere to the ocean increases (Rahmstorf and Willebrand, 1995; Marotzke, 1996). This effectively doubles the heat flux perturbation in that region (although it is small in the global mean). Consequently the AMOC weakening in faf-heat is larger than the expected response for 1pctCO2. However, the net extra heat input to the North Atlantic, including the feedback, is similar in all the models, indicating that the model spread in AMOC weakening in faf-heat is mainly due to
differences in ocean model response, rather than to a spread in the buoyancy forcing.

Despite its exaggerated magnitude, this coupled feedback is a physical effect which must also occur in the CMIP5 1pctCO2 experiments from which the heat flux perturbation was derived (Winton et al., 2013), and presumably in general in climate change simulated by AOGCMs. Our results therefore strongly suggest that it is an important effect on the weakening of the AMOC in response to $CO_2$ forcing, and may not have been sufficiently appreciated. Stammer et al. (2011) found a similar
large positive feedback from increased heat input on the weakening of the AMOC in response to addition of freshwater around Greenland.

Global-mean surface air temperature cools over a large fraction of the world in faf-water, by 0.3 K in the global model mean. The global-mean input by the water flux perturbation is very small compared with its local values (its ocean area-mean





is $7.2 \times 10^{-8} \, \mathrm{kg \, m^{-2} \, s^{-1}}$, two orders of magnitude smaller than its spatial standard deviation of $5.4 \times 10^{-6} \, \mathrm{kg \, m^{-2} \, s^{-1}}$), so

the phenomenon may be a response to its geographical pattern. Global-mean surface temperature change is small in faf-stress and faf-heat (note that the heat added to the ocean in faf-heat is prevented from directly affecting the surface air temperature), but there is substantial warming near to Antarctica and in the Arctic, and strong cooling in the North Atlantic and northern mid-latitude land areas in faf-heat associated with the AMOC weakening. Heat is added in faf-heat mainly at high latitude, and is transported equatorward and downward in a model-dependent way. This implies a spread in ocean heat uptake efficiency and

global-mean sea-level rise due to thermal expansion.

As in many previous studies, the main geographical features of sea-level change are an increase in the gradient across the ACC (small sea-level rise to the south, large to the north), a dipole of sea-level change in the North Atlantic (small sea-level rise in the subtropical gyre, large sea-level rise to the north), and enhanced sea-level rise in the Arctic. We find that the Southern Ocean feature is caused in roughly equal measure by momentum and heat flux perturbations, and somewhat counteracted by

the water flux perturbation. The Arctic feature is mainly due to the water flux perturbation. The North Atlantic feature results from the heat and water flux perturbations, which both give a meridional contrast in buoyancy flux (greater to the north, causing more sea-level rise). In faf-heat this effect is opposed by reduced heat transport due to the weakening of the AMOC, which redistributes heat from high to low latitude. Redistribution is also responsible for strongly enhanced sea-level rise along the Atlantic coast of North America, in the western boundary current. In the Southern Ocean, where there is the greatest increase

in ocean heat content, heat uptake is largely passive.

The results from the pre-CMIP6 trial experiments shows that there will be many qualitative and quantitative features to be analysed in CMIP6. The CMIP6 FAFMIP experiments and the piControl and idealised $CO_2$ experiments with FAFMIP models, will contain process-based diagnostics for rates of change of temperature and salinity due to separate ocean interior transport processes (advection, diffusion, etc.). Such diagnostics have been available in only a few models previously, and were not

included in the preliminary experiments that we have carried out for this paper. They will yield a great deal of new information. In the piControl the diagnostics will enable us to study the balance of ocean processes in the mean state and unforced variability of the coupled atmosphere–ocean system. In the FAFMIP experiments and the idealised $CO_2$ climate-change experiments they will allow us to identify the mechanistic explanations both for the common features of the model responses to surface flux forcing and for the differences among models.

The FAFMIP steering committee will promote the analysis of the experiments, bearing in mind the scientific questions which motivated the project. Comparison of the results from different AOGCMs will aim to identify the causes of the spread in their simulated climate change, in terms of model formulation and emergent behaviour. We envisage that in the light of further analysis we may devise additional tier-2 experiments, for instance to study the effect of surface heat flux feedbacks. It may also be useful to carry out ensemble experiments to quantify the influence of unforced variability, although the major features

of the forced response are expected to be robust in view of the large size of the perturbations. The effect of combining the flux perturbations, in the faf-all experiment, will be studied.

The application of common surface flux perturbations is a technique which has not been widely used up to now as a means to study ocean climate change simulated by AOGCMs in response to $CO_2$ forcing. We therefore hope that the FAFMIP ex-



periments will offer new insight into the reasons for model spread in the ocean response, without the confounding influence
of diversity in atmospheric response. Where the patterns of ocean climate change differ among the models in FAFMIP experiments, we expect that these differences will correspond to those which the models exhibit in the AOGCM scenario-forced
projections. On the other hand, when the ocean models agree in FAFMIP experiments, it will give us greater confidence in the
results, and we will be able to infer that the atmosphere models are the source of uncertainty in projections of ocean climate
change.

**Data availability**

The model output from the DECK and CMIP6 historical simulations described in this paper will be distributed through the
Earth System Grid Federation (ESGF) with digital object identifiers (DOIs) assigned. As in CMIP5, the model output will be
freely accessible through data portals after registration. In order to document CMIP6's scientific impact and enable ongoing
support of CMIP, users are obligated to acknowledge CMIP6, the participating modelling groups, and the ESGF centres (see
details on the CMIP Panel website at http://www.wcrp-climate.org/index.php/wgcm-cmip/about-cmip).

Further information about the infrastructure supporting CMIP6, the metadata describing the model output, and the terms
governing its use are provided by the WGCM Infrastructure Panel (WIP) in their invited contribution to this Special Issue
(Balaji et al., in preparation). Along with the data itself, the provenance of the data will be recorded, and DOIs will be assigned
to collections of output so that they can be appropriately cited. This information will be made readily available so that published
research results can be verified and credit can be given to the modelling groups providing the data. The WIP is coordinating
and encouraging the development of the infrastructure needed to archive and deliver this information. In order to run the
experiments, datasets for natural and anthropogenic forcings are required. These forcing datasets are described in separate
invited contributions to this Special Issue. The forcing datasets will be made available through the ESGF with version control
and DOIs assigned.

*Acknowledgements.* We acknowledge helpful comments during discussions about the design made by John Church, Gokhan Danabasoglu,
Catia Domingues, Till Kuhlbrodt, Jaime Palter, Tatsuo Suzuki and Xuebin Zhang, useful conversations with Chris Roberts and Matt Palmer,
and constructive reviews from Ron Stouffer and Adele Morrison. The model output from the simulations described in this paper will be
distributed through the Earth System Grid Federation with digital object identifiers (DOIs) assigned, and will be freely accessible through
data portals after registration. Further information about the infrastructure supporting CMIP6, the metadata describing the model output, and
the terms governing its use are provided by the WGCM Infrastructure Panel in their invited contribution to this Special Issue.





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



**Table 1.** FAFMIP experiments

| Name | Ocean surface flux perturbation |
| --- | --- |
| Tier 1 | |
| faf-stress | Zonal and meridional momentum |
| faf-heat | Heat |
| faf-water | Freshwater |
| Tier 2 | |
| faf-all | All from faf-stress, faf-heat and faf-water |
| faf-passiveheat | Heat as in faf-heat, but added as a passive tracer |

The process-based tendency diagnostics (Sect. 2.6) should be included in the FAFMIP experiments and in the DECK abrupt4xCO2 and 1pctCO2 and the corresponding section of piControl. The faf-passiveheat experiment is identical to piControl except for the inclusion of the added heat tracer, so a separate integration may not be needed.

**Table 2.** AOGCMs used for FAFMIP preliminary experiments.

| Name | L | Ocean horizontal grid | References | $F$ | $\Delta Q$ | | $\Delta$AMOC | |
| --- | --- | --- | --- | --- | --- | --- | --- | --- |
| | | | | | G | NA | FAF | 1pct |
| HadCM3 | 20 | 1.25° longitude–latitude | Gordon et al. (2000) | 1.80 | −0.21 | 0.42 | −5.6 | −2.3 |
| CanESM2 | 40 | 1.4° longitude × ∼0.93° latitude | Yang and Saenko (2012) with small updates[a] | 1.90 | 0.05 | 0.53 | −7.3 | −2.5 |
| GFDL-ESM2M | 50 | 1° tripolar, refined at low latitude to $\frac{1}{3}$° in tropics | Dunne et al. (2012) | 1.86 | 0.10 | 0.45 | −12.0 | −6.8 |
| MPI-ESM-LR | 40 | 0.13°–1.65° curvilinear | Giorgetta et al. (2013) with small updates | 1.97 | 0.15 | 0.51 | −9.5 | −3.8 |
| GISS-E2-R-CC | 32 | 1.25° longitude × 1.0° latitude | Schmidt et al. (2006) | | | | | |

[a]The most important update is the use of a baroclinicity-dependent formulation for the eddy transfer coefficient in the scheme of Gent and McWilliams (1990).

The column marked "L" indicates the number of ocean model levels. The column marked "F" is the ocean area mean surface heat flux perturbation (W m$^{-2}$) in faf-heat. The columns marked "$\Delta Q$" indicate the time-mean difference in the surface heat flux (W m$^{-2}$) computed by the AOGCM between faf-heat and the control, "G" for the global-mean ocean area, "NA" for the North Atlantic area marked on Fig. 2b. The columns marked "$\Delta$AMOC" indicate the change in the AMOC (Sv), "FAF" for the time-mean of the last decade of faf-heat compared with its control, "1pctCO2" for the time-mean of years 61–80 in 1pctCO2 compared with piControl in CMIP5 results with the same model.



**Table 3.** Ocean model diagnostics of particular interest to FAFMIP analyses (as well as the process-based diagnostics of Table 4).

| CMIP short name | unit | CF standard name |
| --- | --- | --- |
| zos | m | sea_surface_height_above_geoid |
| zostoga | m | global_average_thermosteric_sea_level_change |
| thetao | degC | sea_water_potential_temperature |
| *bigthetao | degC | sea_water_conservative_temperature |
| thetaoga | degC | (volume-mean of thetao) |
| *bigthetaoga | degC | (volume-mean of bigthetao) |
| *opottempmint | degC kg m-2 | integral_wrt_depth_of_product_of_sea_water_density_and_potential_temperature |
| *ocontempmint | degC kg m-2 | integral_wrt_depth_of_product_of_sea_water_density_and_conservative_temperature |
| so | 1e-3 | sea_water_salinity |
| *somint | 1e-3 kg m-2 | integral_wrt_depth_of_product_of_sea_water_density_and_salinity |
| msftmyz | kg s-1 | ocean_meridional_overturning_mass_streamfunction |
| msftyyz | kg s-1 | ocean_y_overturning_mass_streamfunction |
| hfds | W m-2 | surface_downward_heat_flux_in_sea_water |
| wfo | kg m-2 s-1 | water_flux_into_sea_water |
| *rsdoabsorb | W m-2 | net_rate_of_absorption_of_shortwave_energy_in_ocean_layer |
| *pathetao | degC | sea_water_additional_potential_temperature |
| *prthetao | degC | sea_water_redistributed_potential_temperature |
| *pabigthetao | degC | sea_water_additional_conservative_temperature |
| *prbigthetao | degC | sea_water_redistributed_conservative_temperature |

*indicates diagnostics which are newly introduced in CMIP6.

The CMIP short names are used by the Climate Model Output Rewriter (CMOR) software and in naming datasets to be submitted to CMIP6.

The CF standard names are defined by the CF metadata convention `www.cfconventions.org`.



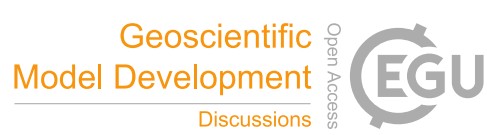

**Table 4.** New diagnostics for process-based ocean temperature and salinity tendencies, required by FAFMIP and described in detail in Sect. 9 of Griffies et al. (submitted).

| CMIP short name | CF standard name |
| --- | --- |
| opottemptend | tendency_of_sea_water_potential_temperature_expressed_as_heat_content |
| opottemprmadvect | tendency_of_sea_water_potential_temperature_expressed_as_heat_content_due_to_residual_mean_advection |
| opottemppadvect | tendency_of_sea_water_potential_temperature_expressed_as_heat_content_due_to_parameterized_eddy_advection |
| opottemppmdiff | tendency_of_sea_water_potential_temperature_expressed_as_heat_content_due_to_parameterized_mesoscale_diffusion |
| opottemppsmadvect | tendency_of_sea_water_potential_temperature_expressed_as_heat_content_due_to_parameterized_submesoscale_advection |
| opottempdiff | tendency_of_sea_water_potential_temperature_expressed_as_heat_content_due_to_parameterized_dianeutral_mixing |
| ocontemptend | tendency_of_sea_water_conservative_temperature_expressed_as_heat_content |
| ocontemprmadvect | tendency_of_sea_water_conservative_temperature_expressed_as_heat_content_due_to_residual_mean_advection |
| ocontemppadvect | tendency_of_sea_water_conservative_temperature_expressed_as_heat_content_due_to_parameterized_eddy_advection |
| ocontemppmdiff | tendency_of_sea_water_conservative_temperature_expressed_as_heat_content_due_to_parameterized_mesoscale_diffusion |
| ocontemppsmadvect | tendency_of_sea_water_conservative_temperature_expressed_as_heat_content_due_to_parameterized_submesoscale_advection |
| ocontempdiff | tendency_of_sea_water_conservative_temperature_expressed_as_heat_content_due_to_parameterized_dianeutral_mixing |
| osalttend | tendency_of_sea_water_salinity_expressed_as_salt_content |
| osaltrmadvect | tendency_of_sea_water_salinity_expressed_as_salt_content_due_to_residual_mean_advection |
| osaltppadvect | tendency_of_sea_water_salinity_expressed_as_salt_content_due_to_parameterized_eddy_advection |
| osaltpmdiff | tendency_of_sea_water_salinity_expressed_as_salt_content_due_to_parameterized_mesoscale_diffusion |
| osaltpsmadvect | tendency_of_sea_water_salinity_expressed_as_salt_content_due_to_parameterized_submesoscale_advection |
| osaltdiff | tendency_of_sea_water_salinity_expressed_as_salt_content_due_to_parameterized_dianeutral_mixing |

The units of the temperature tendency diagnostics are $W\,m^{-2}$, and of the salinity tendency diagnostics $kg\,m^{-2}\,s^{-1}$. Either the potential temperature or the conservative temperature diagnostics should be included, depending on which is the prognostic of the model. The effect of advection by the model (resolved) velocity field can be calculated as the difference between the effects of residual mean advection and parameterized eddy advection. The latter should include both mesoscale and submesoscale effects. If these are not distinguished in the model, the diagnostics for parameterized submesoscale advection should be omitted.