# Peer review of "The Flux-Anomaly-Forced Model Intercomparison Project (FAFMIP) contribution to CMIP6: Investigation of sea-level and ocean climate change in response to CO2 forcing"

_Geoscientific Model Development, 2016_

## Referee Comment (RC1) · Anonymous Referee #1 · 6 Jul 2016

This manuscript documents the protocol of the Flux-Anomaly-Forced Model Intercomparison Project (FAFMIP) and presents the results obtained so far. As a component of the forthcoming CMIP6, the goal of FAFMIP is to investigate the spread of the fully coupled AOGCMs in simulating sea level rise and ocean climate change under CO2 forcing, especially the roles of momentum, heat and water fluxes at the ocean surface. It is found that CO2-induced heat flux anomalies are the dominant factor in causing the weakening of the Atlantic meridional overturning circulation (AMOC) and the dipole

pattern of dynamic sea level changes in the North Atlantic. In the Southern Ocean, both momentum and heat fluxes contribute to the increased dynamic sea level gradient across the Antarctic Circumpolar Current. Different processes responsible for the ocean heat content change, such as added and redistributed heat, are also investigated and compared.

This is an important manuscript that represents a community effort. This type of manuscript systematically describes the design of coordinated projects, provides guidance for follow-up studies, and usually gets frequent citations. For the climate modeling community, a better understanding of model spread is as important as the identification of common simulation features across models. In particular, momentum, heat and water fluxes at the ocean surface are tightly coupled. It is not easy to separate their individual contributions to ocean climate change and sea level rise pattern. By reading the manuscript, I found the design of FAFMIP is clever, reasonable, and quite effective at revealing the role of different air-sea fluxes. Although compared with previous studies, no significant new conclusion has been drawn so far with FAFMIP, the project tackles critical issues in a more systematic and complete way, and provides the community with valuable data for further analyses. For example, the proposed new ocean diagnostics would facilitate attribution studies. The outcome would benefit the climate modeling community.

The manuscript is well written. So I recommend publication of the manuscript in Geoscientific Model Development. My comments below are minor and mainly for clarification purpose.

(1) Line 35: Please define "YJ" as some readers may not be familiar with this unit.

(2) Lines 199-200: Is the purpose of the faf-all experiments to compare with the original coupled simulations? The motivation for faf-all needs further discussion.

(3) Section 2.4: In this section, three methods for treating the surface heat fluxes are discussed. To help the understanding, I suggest adding a schematic illustration for

better demonstration and comparison of the three methods.

(4) Figure 3: In Panel b, the global mean surface air temperature is relatively stable in Method B compared with Method A. In Panel d, by contrast, the ocean volume-mean temperature increases faster for Method B than Method A. Looking into the text, the reason is such that SST is determined by TR without the influence of the heat flux perturbation - F. By contrast, temperatures in the ocean interior are affected by F. This difference can be reiterated in the caption of Fig. 3 as readers may first look at figures before digging into the text.

(5) Lines 247-248: The weakening of the AMOC causes a cooling in the northern North Atlantic and an increase in heat flux into the ocean. Does that actually mean a decrease in oceanic heat loss in this region? The effect on temperature would be the same but physically different.

(6) Line 334: Please define DECK.

(7) Line 400: Heat flux perturbations -> Heat flux anomalies. The former implies external while the latter can be internal.

(8) Lines 430-435: This paragraph discusses $\sigma\zeta$ which contains both spatial and temporal variability of $\zeta$. From the bottom row of Fig. 4, any information can be inferred about the "signal-to-noise" ratio or the time of emergence of the externally forced signals?

(9) Line 445: The Pacific Decadal Oscillation plays an important role in causing the largest regional sea level trends in the Pacific since the 1990s. Without ocean initialization, models are unlikely to capture these trends on decadal/interdecadal time scales. These comments could be added regarding model performance.

(10) Lines 551-555: The positive feedback between the AMOC-induced cooling and the increase in downward heat flux into the ocean is interesting. As I know, the temperature feedback associated with AMOC can also operate in another way. The

[Figure]

AMOC-induced ocean cooling causes increase in surface water density, which in turn enhances oceanic deep convection and strengthens AMOC. I'd like to see some discussion about how to reconcile these feedback processes.

———————————————

---

## Referee Comment (RC2) · Anonymous Referee #2 · 28 Jul 2016

This manuscript outlines the formulation of the FAFMIP experiments – a model inter-comparison in which air-sea fluxes a forced by different scenarios. The goal of the experiments are to reveal the extent to which the ocean model controls the climate response to differing scenarios. This information can then be used to guide interpretation of standard CMIP simulations, and to attribute observed changes to individual forcing mechanisms.

This is a very well-written paper, and it's clear that a lot of thought has gone into the

design of FAFMIP. The method used to conduct the heat flux experiments are relatively convoluted due to unavoidable feedback effects, but both the issues faced and solutions proposed were clearly explained in the manuscript. In short, I am happy to recommend publication of this paper without alteration, and have only minor suggestions that the authors may want to consider.

1. Around line 180 the authors specify that momentum stress should be added only to the momentum balance, and should not alter mixed layer turbulence schemes. A little more information on how to do this (e.g. in the form of equations, separating the perturbation stress from the unperturbed stress, as is done for the far-heat experiments) would be helpful.

2. The "added heat" tracer was introduced rather suddenly on line 310, without a definition. It quickly became obvious what it is intended to represent, but a simple equation relating $T_A$ to $\theta$ would have made this clearer.

3. In the last paragraph of p.16 there was a statement about heat being more influential on the AMOC than freshwater. It's worthwhile comparing the magnitude of buoyancy flux perturbations in each case – in particular, is this result purely because the buoyancy fluxes are greater in the heat case, or is the feedback on heat more significant?

4. Line 457 - This sentence didn't quite make sense; I think there may be a missing word.

5. My only significant criticism of this paper is that the "Preliminary Results" section is very hard to read, as it jumps back and forth between different figures, and doesn't seem to have a clear message. This may be partly because the results are indeed preliminary, but (for example) creating subsections covering the AMOC, OHC and SLR, and reformatting the figures accordingly, would help the readability of the paper.

---

## Author Comment (AC1) · 30 Sep 2016

We are grateful for the referees' time, and encouraged by their positive comments, to which our responses are below.

[Figure]

Referee 1

[1.] *Line 35: Please define "YJ" as some readers may not be familiar with this unit.*

We will note that 1 YJ $\equiv 10^{24}$ J.

[2.] *Lines 199-200: Is the purpose of the faf-all experiments to compare with the original coupled simulations? The motivation for faf-all needs further discussion.*

By comparison with the tier-1 experiments, faf-all will be used to quantify non-linearities in the combination of the effects of the perturbations. The faf-all experiments will not be quantitatively the same as the coupled 1pctCO2 experiments, because the flux perturbations are a model mean with no time-dependence. However, we expect they will be qualitatively similar, so if faf-all suggests that the combination is linear, the ocean response to $CO_2$ forcing in coupled simulations may be interpreted as the sum of the effects. This should be noted where faf-all is introduced.

[3.] *Section 2.4: In this section, three methods for treating the surface heat fluxes are discussed. To help the understanding, I suggest adding a schematic illustration for better demonstration and comparison of the three methods.*

That is a good idea. We include a possible figure at the end of this document.

[4.] *Figure 3: In Panel b, the global mean surface air temperature is relatively stable in Method B compared with Method A. In Panel d, by contrast, the ocean volume-mean temperature increases faster for Method B than Method A. Looking into the text, the reason is such that SST is determined by TR without the influence of the heat flux perturbation $F$. By contrast, temperatures in the ocean interior are affected by $F$. This difference can be reiterated in the caption of Fig. 3 as readers may first look at figures before digging into the text.*

There would not be space to repeat the explanation in detail, but it can be summarised by noting that in Method B the heat flux perturbation $F$ causes $\Delta T$ to increase, giving

a negative feedback on ocean heat uptake, whereas this is prevented in the other methods, and referring to Sect. 2.4 for details.

[5.] *Lines 247-248: The weakening of the AMOC causes a cooling in the northern North Atlantic and an increase in heat flux into the ocean. Does that actually mean a decrease in oceanic heat loss in this region? The effect on temperature would be the same but physically different.*

In the majority of the North Atlantic there is still a net heat loss by the ocean. We will note this to avoid confusion.

[6.] *Line 334: Please define DECK.*

The CMIP6 DECK is a small set of experiments (including piControl and 1pctCO2) used to evaluate model characteristics of climate and climate change (Eyring , 2016). This should be stated when the DECK is first mentioned.

[7.] *Line 400: Heat flux perturbations → Heat flux anomalies. The former implies external while the latter can be internal.*

That is a good point. We prefer "heat flux variations".

[8.] *Lines 430-435: This paragraph discusses $\sigma_\zeta$ which contains both spatial and temporal variability of $\zeta$. From the bottom row of Fig. 4, any information can be inferred about the "signal-to-noise" ratio or the time of emergence of the externally forced signals?*

Thank you for making this point. Yes, in all models $\sigma_\zeta$ in faf-heat becomes significantly different from the control early in the experiment, implying that a statistically detectable geographical pattern of forced change in dynamic sea-level has emerged from the background of unforced variability. This idea is related to the global time of emergence, evaluated by Bilbao (2015) using correlation coefficients. To illustrate it, we propose to indicate statistically significant change in Fig. 4 (modified version at the end of this document) and discuss it in the text, both for $\sigma_\zeta$ and for the AMOC.

[9.] *Line 445: The Pacific Decadal Oscillation plays an important role in causing the largest regional sea level trends in the Pacific since the 1990s. Without ocean initialization, models are unlikely to capture these trends on decadal/interdecadal time scales. These comments could be added regarding model performance.*

We would remark that the east–west contrast in the Pacific, which is not a pattern predicted in response to $CO_2$ forcing by AOGCMs (Bilbao , 2015; Palanisamy , 2015), may partly be due to unforced multiannual variability associated with the Pacific Decadal Oscillation and the Southern Oscillation (Merrifield , 2012; Zhang and Church, 2012). As the referee notes, this will not be reproduced in AOGCM simulations without initialization to an observed state. Even then it would not be reproduced over such a long period, because predictability is limited (Roberts , in press). Moreover, the observed trends have much greater magnitude than spontaneously generated in AOGCM control experiments. A satisfactory explanation for the observed trends is currently lacking (Clark , 2015).

[10.] *Lines 551-555: The positive feedback between the AMOC-induced cooling and the increase in downward heat flux into the ocean is interesting. As I know, the temperature feedback associated with AMOC can also operate in another way. The AMOC-induced ocean cooling causes increase in surface water density, which in turn enhances oceanic deep convection and strengthens AMOC. I'd like to see some discussion about how to reconcile these feedback processes.*

We agree that it would be useful to mention this effect as well, in order to clarify the summary. We would note that the positive feedback on AMOC weakening from the increased heat input is a distinct effect from the negative feedback on AMOC weakening in which the cooling in the North Atlantic promotes convection and deep water formation and tends to strengthen the circulation (e.g. Marotzke, 1996). This negative feedback is an oceanic phenomenon, not a coupled one.

[Figure]

Referee 2

[1.] *Around line 180 the authors specify that momentum stress should be added only to the momentum balance, and should not alter mixed layer turbulence schemes. A little more information on how to do this ( in the form of equations, separating the perturbation stress from the unperturbed stress, as is done for the faf-heat experiments) would be helpful.*

With the stress perturbation imposed, the equation of motion of the top layer of a hydrostatic ocean model is

$$\frac{\partial \mathbf{u}_h}{\partial t} = -(\mathbf{u} \cdot \boldsymbol{\nabla})\mathbf{u}_h - \frac{1}{\rho}\boldsymbol{\nabla}_h p - \mathbf{f} \times \mathbf{u}_h + \frac{1}{\rho}\mathbf{R} + \frac{1}{m_t}(\boldsymbol{\tau}_w + \boldsymbol{\tau}_i + \mathbf{S}),$$

where $\mathbf{u}$ is velocity, subscript $h$ indicates the horizontal part, $t$ is time, $p$ hydrostatic pressure, $\rho$ density, $\mathbf{f}$ the product of the Coriolis parameter and the vertical unit vector, $\mathbf{R}$ the vertical and horizontal convergence of horizontal momentum (in $\mathrm{N\,m^{-3}}$) due to subgridscale processes (including the shear stress which conveys the surface momentum fluxes into the subsurface), $\boldsymbol{\tau}_w$ the windstress, $\boldsymbol{\tau}_i$ the stress exerted by sea-ice, $\mathbf{S}$ the faf-stress momentum flux perturbation (in Pa, like $\boldsymbol{\tau}_{w,i}$), and $m_t$ is the mass per unit area of the top layer of the model, to which the surface momentum fluxes are applied. No perturbation should be made directly to any turbulent mixing scheme that depends on the windstress, nor to the sea-ice momentum balance, although both of these could be indirectly influenced since $\boldsymbol{\tau}_w$ and $\boldsymbol{\tau}_i$ may be affected by changes in the surface $\mathbf{u}_h$.

[2.] *The "added heat" tracer was introduced rather suddenly on line 310, without a definition. It quickly became obvious what it is intended to represent, but a simple equation relating $T_A$ to $\theta$ would have made this clearer.*

We suggest that this could be clarified by reordering the text to state the purpose of $T_A$ at the outset, thus: In order to reveal where the extra heat from the heat flux perturbation is stored in the ocean, we include an "added heat" tracer $T_A$ in faf-heat.

This tracer is initialised to zero (so $\Delta T_A \equiv T_A$) and it has $F$ as its surface flux (we note that heat is added in the global mean, although $F$ is not positive everywhere, as seen in Fig. 2b).

[3.] *In the last paragraph of p. 16 there was a statement about heat being more influential on the AMOC than freshwater. It's worthwhile comparing the magnitude of buoyancy flux perturbations in each case—in particular, is this result purely because the buoyancy fluxes are greater in the heat case, or is the feedback on heat more significant?*

The additional buoyancy flux into the North Atlantic due to the heat flux perturbation is greater by a factor of more than 40 than that due to the water flux perturbation. We propose to remark on this.

[4.] *Line 457. This sentence didn't quite make sense; I think there may be a missing word.*

In fact there was an extra word! The sentence should read "In the Arctic there is reduced OHC and $\Delta\zeta$ is predominantly halosteric ..."

[5.] *My only significant criticism of this paper is that the "Preliminary Results" section is very hard to read, as it jumps back and forth between different figures, and doesn't seem to have a clear message. This may be partly because the results are indeed preliminary, but (for example) creating subsections covering the AMOC, OHC and SLR, and reformatting the figures accordingly, would help the readability of the paper.*

Thanks for this comment. To address it, we would rearrange of Sects. 3.1 (on time-dependence of change) and 3.2 (on spatial patterns of change) to separate more neatly the quantities considered, namely surface air temperature, ocean temperature, AMOC and dynamic sea-level, with subheadings for each quantity. To be more informative, and for consistency, we would like to include the timeseries of global-mean surface air temperature change in the figure with the other timeseries (Fig. 4 in the submitted

manuscript, new version at the end of this document). The rearrangement changes the order of the figures, such that the text refers almost only to adjacently numbered figures at any point, and less jumping around is needed. To aid comparison, we propose to repeat the panels for ocean temperature change (vertical profile, map of vertical integral, zonal-mean cross-section) in the figure which shows the changes in added and redistributed heat tracers (Fig. 9 in the submitted manuscript, new version at the end of this document).

Other comments

We have made minor corrections and changes suggested by informal reviews kindly provided to us by our colleagues Ron Stouffer and Adele Morrison. In response to their comments, we propose to:

[1.] Insert an introductory paragraph to Sect. 2 and make a small change in Sect. 1 to clarify the intention of the design of FAFMIP, especially the tier-1 experiments.

[2.] Emphasise the role of wind-driven redistribution of heat in faf-stress.

We also intend to make various minor clarifications and corrections noted by the authors. Since the initial submission, the faf-heat experiment in HadCM3 has been rerun with absorption of solar radiation having the same vertical profile for $\theta$ and $T_R$, as recommended in the paper. This does not affect any of the qualitative findings of the paper and makes only minor quantitative differences, as a consequence of which the results of HadCM3 have become more similar to those of the other models.

For reference, we attach a revised manuscript that we have prepared as a supplement to this document.

Captions for new and revised figures

**Fig. 1—New figure in revised paper.** The three methods for treating the surface heat flux in faf-heat and faf-all, described in Sect. 2.4. The methods differ regarding the SST which is used to calculate the net surface heat flux $Q$ from the atmosphere and sea-ice into the ocean water. In Method A it is obtained from the top-layer ocean temperature $\theta$ as usual in an AOGCM, in Method B from the redistributed heat tracer $T_R$ and in Method C the SST and sea-ice are prescribed from the climatology of the AOGCM control experiment.

**Fig. 2—Revised version of Fig. 4 of discussion paper.** Annual timeseries in faf-stress, faf-heat and faf-water, according to the key in the first panel. Top row, global-mean surface air temperature change (K) with respect to the control time-mean; second row, ocean volume-mean temperature change (K) with respect to the corresponding year of the control; third row, maximum of the Atlantic meridional overturning stream-function (Sv); bottom row: $\sigma_\zeta$ (m), the spatial standard deviation of $\Delta\zeta$, the dynamic sea-level change relative to the 70-year time-mean in the control experiment. For the AMOC and $\Delta\zeta$, the grey band indicates the range of values which do not differ significantly (as defined in the text) from the control time-mean, which is indicated by the dotted line. *This figure will be in landscape format.*

**Fig. 3—Revised version of Fig. 9 of discussion paper.** Change in ocean temperature $\theta$ (left) and tracers of added heat (centre) and redistributed heat (right) in the final decade of the faf-heat experiment, (top) global-mean change in tracer (K) as a function of depth, with different scales for the temperature axis, (middle) model-mean change in heat content (GJ m$^{-2}$, the vertical integral of the change in tracer multiplied by volumetric heat capacity), (bottom) model-mean zonal-mean cross-sections of the change in tracer (K). *We will note that (a,d,g) repeat earlier figures, and give their numbers, which will have changed from the submitted manuscript because of reordering Sect. 3. This figure will be in landscape format.*

Please also note the supplement to this comment:
http://www.geosci-model-dev-discuss.net/gmd-2016-122/gmd-2016-122-AC1-supplement.pdf

―――――――――――――――――

[Figure]

[Figure]

[Figure]

**Fig. 1.** New figure in revised paper.

[Figure]

**Fig. 2.** Revised version of Fig. 4 of discussion paper.

**(a) Δθ**

**(b) added ΔT$_A$**

**(c) redistributed ΔT$_R$**

HadCM3
CanESM2
GFDL-ESM2M
MPI-ESM-LR

**(d) model-mean ΔOHC**

**(e) model-mean added ΔOHC**

**(f) model-mean redistributed ΔOHC**

**(g) model-mean Δθ**

**(h) model-mean added ΔT$_A$**

**(i) model-mean redistributed ΔT$_R$**

**Fig. 3.** Revised version of Fig. 9 of discussion paper.

**Supplement:**

[revised manuscript text omitted]

---

## Author Comment (AC2) · 30 Sep 2016

Please see the response to referee 1, which contains the responses to all comments